# OpenReview forum: "Entropy Voting Between Capsules"
_TMLR — Rejected by TMLR_

### Review · Reviewer_Cxad · 2025-03-19

**Summary Of Contributions:**

In this work, the authors propose a theoretical study to leverage the generalizability of capsule network architectures. More specifically, through their approach, the main goal would be to maximize at the same time information extracted by the capsules and maximize the agreement obtained through the routing mechanism. This is modeled by interpreting every capsule as a random variable, which results in a log-normal marginal differential entropy distribution. The agreement between parent and child capsules is evaluated through a KL divergence. The experiments are conducted on MNIST and SmallNORB.

The main contribution relies upon the entropy-voting mechanism.

**Audience:**

Yes

**Claims And Evidence:**

No

**Requested Changes:**

- [Critical] Please provide technical and theoretical depth to the work. As it stands, it is unclear why the proposed approach should be better than existing routing mechanisms, and why it should make the training less sensitive to augmentations.

- [Critical] Please enlarge the experimental setup with more complex datasets.

- [Critical] Please apply the routing mechanism to existing architectures, showing a clear gain led by the proposed routing approach.

- [Important] Please provide implementation details, sensitivity analysis, and possibly the code for reproducibility.

- [Important] Please update the literature referenced in the paper, comparing with other entropy-based approaches applied to capsules and clearly differentiating from them.

- [Less important] Please include more figures to describe your method, and to provide an overview on the advantages of your approach.

**Strengths And Weaknesses:**

*Strength*
- The paper is in general easy to follow and in most of its parts, logical. The concept of modeling agreement between capsules through an entropic estimator is certainly one possibility.

*Weaknesses*
- Technical depth. Although the work flows, it is presented as-is, without analyzing the theoretical properties of the proposed approach - what are the nice properties deriving from the proposed approach ? Is there a direct link with capsule's generalizability? Why (whether that is the case) should it make capsule's training less requiring augmentations (as it is celebrated in the experimental section)? As it stands, there is no real motivation (aside from looking at pure performance) or technical depth in the work.

- Very limited experiments. Testing only on MNIST and SmallNORB, on one capsule architecture, is largely insufficient. The authors should include at least more complex datasets like FashionMNIST, CIFAR-10, and ImageNet.

- Performance comparison is confusing. Although the attempt to report performance from other works in Table 2 is appreciated, it is unclear how to really compare with the method's result provided in Table 1. It would be more useful to reimplement those approaches and to test them directly without augmentations rather than reporting their numbers.

- Augmentation harms performance in SmallNORB. Although it is clear that the authors would like to focus on no augmentation setup, this conceptually does not find a straightforward explanation: providing explicitly a prior (the augmentation) should not change the performance of the trained model if the proposed approach was robust to the same transformation set by-design. This raises several questions on the validity of the proposed approach for the indicated purposes.

- Comparison between architectures is not fair. Overall, the proposed approach is not really "architectural" and the same routing mechanism should be applicable to any of the other proposed approaches. Proposing a different architecture makes it hard to evaluate the merits of the routing mechanism being proposed. Also, it looks like from Alg.1 that one iteration only is conducted for the proposed approach - are the other methods evaluated under the same conditions?

- Related works and reference literature is outdated. The latest referenced work is dated 2021, but since then there have been multiple efforts in the domain, including entropy estimators applied to capsules. Just to name a few [A-E].

- Reproducibility. One of the major issues in Capsule's literature is the (ir)reproducibility of some results. Although the authors provide the list of hyperparameters, it is unclear how they are found, and how sensitive the performance is to their tuning.

[A] Renzulli, Riccardo, et al. "Rem: Routing entropy minimization for capsule networks." arXiv preprint arXiv:2204.01298 (2022).

[B] Liu, Yi, et al. "Capsule networks with residual pose routing." IEEE Transactions on Neural Networks and Learning Systems (2024).

[C] Liu, Xiaoyang, et al. "Link prediction approach combined graph neural network with capsule network." Expert Systems with Applications 212 (2023): 118737.

[D] Zhou, Heng, et al. "Image classification based on quaternion-valued capsule network." Applied Intelligence 53.5 (2023): 5587-5606.

[E] Yu, Chang, et al. "HP-capsule: Unsupervised face part discovery by hierarchical parsing capsule network." Proceedings of the IEEE/CVF Conference on Computer Vision and Pattern Recognition. 2022.

---

### Review · Reviewer_GJat · 2025-03-21

**Summary Of Contributions:**

This paper proposes a new routing algorithm for capsule networks (CapsNets) based on information theory. This method computes marginal differential entropy terms and KL divergence between child and parent capsules. Experiments are conducted on MNIST and smallNORB datasets.

**Audience:**

Yes

**Broader Impact Concerns:**

No concerns

**Claims And Evidence:**

No

**Requested Changes:**

See weaknesses above.

**Strengths And Weaknesses:**

### Strengths

- The proposed routing algorithm is non-iterative.

### Weaknesses
I have many concers on this paper, starting from the methodology part to experimental settings.

- The notation should be improved. In all Equations, please use capsule notation for consistency.

- I don't get how you compute the entropies and KL divergences. Can you provide the exact formulas? It's not clear also because the number of child and parent capsules could be different, and typically the routing algorithm is performed between votes of child capsules to parent capsules and parent capsules, not between child capsules and parent capsules.

- From Figure 2, the network is basically a convolutional network. The authors should compare with a network with more capsule layers (so more routing layers), number of capsules in each layer and their dimensions to effectively show the advantages of the proposed novel algorithm.

- Are the coupling coefficients computed? Additional plots or visualizations of their distributions should be included.

- The experimental setup is very limited. I know that CapsNets still struggles with high resolution datasets and many output classes, but authors should perform additional experiments at least on cifar10 and SVHN. Furthermore, the standar setting of smallNORB is to test on different viewpoints (azimuths and elevations).

- Some references are missing. For example "REM: Routing Entropy Minimization for Capsule Networks, Riccardo Renzulli, Enzo Tartaglione, Marco Grangetto" should be mentioned in the related work since it focuses on entropy too.

---

### Review · Reviewer_EKK6 · 2025-03-23

**Summary Of Contributions:**

This paper proposes an information-theoretic based voting mechanism for capsule networks, which aims to improve the performance of capsule networks by maximizing the marginal entropy of the capsules and minimizing the relative entropy between low-level and high-level capsules. The motivation of the paper is clear, the method is novel, and the experimental results show that the method outperforms existing capsule networks on MNIST and smallNORB datasets, especially in the inference speed.

**Audience:**

Yes

**Broader Impact Concerns:**

n/a.

**Claims And Evidence:**

Yes

**Requested Changes:**

1.Insufficient theoretical depth: Add a theoretical analysis of the combination of information theory and capsule networks to explain why maximizing marginal entropy and minimizing KL dispersion can improve the performance of capsule networks. More relevant literature could be cited to support this idea.
2.Experimental Limitations: the paper only conducted experiments on two relatively simple datasets, MNIST and smallNORB, and lacks validation on more complex datasets such as CIFAR-10. This limits the generalizability of the paper's conclusions.
3.Inadequate comparisons: the paper only compares with three capsule network baselines (Sabour et al., Mazzia et al., and Byerly et al.'s methods), and does not compare with other broader methods.
4.The cited papers are too old: much of the literature cited in the paper focuses on 2017-2019, while the fields of capsule networks and computer vision have made significant progress in recent years (2020-2025). Many of the most recent research results are not cited. It is recommended to cite some research results from recent years, for example:
[1] Yang Yang, Qi Qin, Yongjiang Luo, Yi Liu*, Qiang Zhang*, and Jungong Han, Bi-directional progressive guidance network for RGB-D salient object detection, IEEE Transactions on Circuits and Systems for Video Technology (T-CSVT), 2022, 32(8): 5346-5360.
[2] Yi Liu, Xiaohui Dong, Dingwen Zhang, Shoukun Xu, Deep unsupervised part-whole relational visual saliency, Neurocomputing, 2024, 563: 126916.
[3] Yi Liu, Chengxin Li, Shoukun Xu, Jungong Han, Part-whole relational fusion towards multi-modal understanding, International Journal of Computer Vision (IJCV).

**Strengths And Weaknesses:**

Strengths: This paper proposes an information-theoretic based voting mechanism for capsule networks, which aims to improve the performance of capsule networks by maximizing the marginal entropy of the capsules and minimizing the relative entropy between low-level and high-level capsules. The motivation of the paper is clear, the method is novel, and the experimental results show that the method outperforms existing capsule networks on MNIST and smallNORB datasets, especially in the inference speed.
Weakness: Please see the changes below.

---

### Public Comment · ~Yi_Liu24 · 2025-03-21

This paper proposes an information-theoretic based voting mechanism for capsule networks, which aims to improve the performance of capsule networks by maximizing the marginal entropy of the capsules and minimizing the relative entropy between low-level and high-level capsules. The motivation of the paper is clear, the method is novel, and the experimental results show that the method outperforms existing capsule networks on MNIST and smallNORB datasets, especially in the inference speed. However, the paper is still deficient in some aspects and needs further improvement to enhance its contribution and readability.
There are a few concerns：
1.Insufficient theoretical depth: Add a theoretical analysis of the combination of information theory and capsule networks to explain why maximizing marginal entropy and minimizing KL dispersion can improve the performance of capsule networks. More relevant literature could be cited to support this idea.
2.Experimental Limitations: the paper only conducted experiments on two relatively simple datasets, MNIST and smallNORB, and lacks validation on more complex datasets such as CIFAR-10. This limits the generalizability of the paper's conclusions.
3.Inadequate comparisons: the paper only compares with three capsule network baselines (Sabour et al., Mazzia et al., and Byerly et al.'s methods), and does not compare with other broader methods.
4.The cited papers are too old: much of the literature cited in the paper focuses on 2017-2019, while the fields of capsule networks and computer vision have made significant progress in recent years (2020-2025). Many of the most recent research results are not cited. It is recommended to cite some research results from recent years, for example:
[1 ]Yang Yang, Qi Qin, Yongjiang Luo, Yi Liu*, Qiang Zhang*, and Jungong Han, Bi-directional progressive guidance network for RGB-D salient object detection, IEEE Transactions on Circuits and Systems for Video Technology (T-CSVT), 2022, 32(8): 5346-5360.
[2] Yi Liu, Xiaohui Dong, Dingwen Zhang, Shoukun Xu, Deep unsupervised part-whole relational [3]visual saliency, Neurocomputing, 2024, 563: 126916. Yi Liu, Chengxin Li, Shoukun Xu, Jungong Han, Part-whole relational fusion towards multi-modal understanding, International Journal of Computer Vision (IJCV).

Major Revision.

---

### Comment · Reviewer_EKK6 · 2025-03-22
**Major Revision**

This paper proposes an information-theoretic based voting mechanism for capsule networks, which aims to improve the performance of capsule networks by maximizing the marginal entropy of the capsules and minimizing the relative entropy between low-level and high-level capsules. The motivation of the paper is clear, the method is novel, and the experimental results show that the method outperforms existing capsule networks on MNIST and smallNORB datasets, especially in the inference speed. However, the paper is still deficient in some aspects and needs further improvement to enhance its contribution and readability. There are a few concerns： 1.Insufficient theoretical depth: Add a theoretical analysis of the combination of information theory and capsule networks to explain why maximizing marginal entropy and minimizing KL dispersion can improve the performance of capsule networks. More relevant literature could be cited to support this idea. 2.Experimental Limitations: the paper only conducted experiments on two relatively simple datasets, MNIST and smallNORB, and lacks validation on more complex datasets such as CIFAR-10. This limits the generalizability of the paper's conclusions. 3.Inadequate comparisons: the paper only compares with three capsule network baselines (Sabour et al., Mazzia et al., and Byerly et al.'s methods), and does not compare with other broader methods. 4.The cited papers are too old: much of the literature cited in the paper focuses on 2017-2019, while the fields of capsule networks and computer vision have made significant progress in recent years (2020-2025). Many of the most recent research results are not cited. It is recommended to cite some research results from recent years, for example: [1 ]Yang Yang, Qi Qin, Yongjiang Luo, Yi Liu*, Qiang Zhang*, and Jungong Han, Bi-directional progressive guidance network for RGB-D salient object detection, IEEE Transactions on Circuits and Systems for Video Technology (T-CSVT), 2022, 32(8): 5346-5360. [2] Yi Liu, Xiaohui Dong, Dingwen Zhang, Shoukun Xu, Deep unsupervised part-whole relational [3]visual saliency, Neurocomputing, 2024, 563: 126916. Yi Liu, Chengxin Li, Shoukun Xu, Jungong Han, Part-whole relational fusion towards multi-modal understanding, International Journal of Computer Vision (IJCV).

---

### Decision · Action_Editor_pZUp · 2025-04-28

**Recommendation:** Reject

**Comment:**

The reviewers have collectively acknowledged some of the novel elements of this method, which allows for improving the routing process of capsule neural networks, making them better performing in some instances and more efficient. From a technical standpoint the method is clearly described and is easy to follow. However, the key limitations and weaknesses that have not been addressed and which affect the claims and evidence provided include the following:

a) Expanding literature, including which methods have been selected to compare against, should have been more extensive, covering more recent papers published post-2020 and even in 2024 and 2025

b) The experimental setup is rather limited; in fact the results on the main dataset of interest, SmallNorb, are inconclusive, especially in the presence of other previous papers that perform very well on this dataset. In fact, I think this submission would have been stronger if more datasets had been included to understand the full capabilities of the proposed method, even if performance had been lower than SOTA. Affnist and SVHN would have been great additions, and even more recent, larger ones such as 3DIEBench.

c) On SmallNorb, it is quite common for papers (even those already cited) to present performance separately for azimuth and elevation (familiar/novel viewpoints). This should have been done in this instance as well, as one of the reviewers suggested.

I encourage the authors to revise and expand their paper, considering the reviewers' feedback, and then contemplate resubmitting it. The method has merit and warrants further exploration and expansion.

**Audience:**

The intended audience for this paper includes researchers in the fields of computer vision and machine learning, particularly those interested in capsule networks and information theory. The paper is relevant to individuals looking to improve the generalisation and efficiency of capsule networks without extensive data augmentation. Capsule neural networks have recently attracted increased attention, with several papers published in major conferences and journals, demonstrating great performance on equivariance-related and other tasks, even in comparison to CNNs and ViTs.

**Claims And Evidence:**

This paper claims that capsule networks can be enhanced by using an entropy voting method based on information theory. The authors argue that by treating capsules as continuous random variables and optimising their marginal differential entropy and relative entropy, the performance of capsule networks can be improved. The evidence supporting these claims includes experimental results demonstrating that the proposed method outperforms (some) existing capsule networks on MNIST and smallNORB datasets, particularly in terms of inference speed. An ablation study included in the paper confirms the relationship between capsule entropies and network performance. However, reviewers have pointed out several weaknesses, such as insufficient theoretical depth, limited experimental validation on complex datasets, and outdated literature citations. Some of these issues have been acknowledged by the authors, but they were not adequately addressed, e.g. via a revised paper submission.

In addition to the requirement to be more current in the capsule literature, the paper's experimental setup is fairly limited compared to what is needed to understand how this paper pushes the boundaries in capsule neural network research. Datasets such as cifar10/100, affnist, fashionmnist, multimnist, SVHN, 3DIEBench or similar could have been utilised to provide a more comprehensive evaluation and, therefore, evidence of the proposed method that would support the claims.

**Resubmission Of Major Revision:**

The authors may consider submitting a major revision at a later time.